# Using Activity Measures and GNSS Data from a Virtual Fencing System to Assess Habitat Preference and Habitat Utilisation Patterns in Cattle

**DOI:** 10.3390/ani14101506

**Published:** 2024-05-19

**Authors:** Magnus Fjord Aaser, Søren Krabbe Staahltoft, Martin Andersen, Aage Kristian Olsen Alstrup, Christian Sonne, Dan Bruhn, John Frikke, Cino Pertoldi

**Affiliations:** 1Department of Chemistry and Bioscience, Aalborg University, Fredrik Bajers Vej 7H, 9220 Aalborg, Denmark; sstaah20@student.aau.dk (S.K.S.); maande16@student.aau.dk (M.A.); db@bio.aau.dk (D.B.); cp@bio.aau.dk (C.P.); 2Department of Nuclear Medicine and PET, Aarhus University Hospital, Palle Juul-Jensens Boulevard 165, 8200 Aarhus, Denmark; aagealst@rm.dk; 3Department of Clinical Medicine, Aarhus University, Palle Juul-Jensens Boulevard 165, 8200 Aarhus, Denmark; 4Department of Ecoscience, Aarhus University, Frederiksborgvej 399, 4000 Roskilde, Denmark; cs@ecos.au.dk; 5Skagen Bird Observatory, Fyrvej 36, 9990 Skagen, Denmark; 6Wadden Sea National Park, Havnebyvej 30, 6792 Rømø, Denmark; jofri@danmarksnationalparker.dk; 7Aalborg Zoo, Mølleparkvej 63, 9000 Aalborg, Denmark

**Keywords:** animals, virtual fencing, grazing management, Nofence©, cattle, habitat preference, habitat utilisation

## Abstract

**Simple Summary:**

Developing sustainable livestock management requires knowledge and monitoring of which habitats within an enclosure the livestock prefers to stay in and in what way they utilise the available habitats. The technology of virtual fencing provides the ability to both monitor and adapt enclosures, thus adding value to the expense related to fencing for farmers and livestock managers. To this end, the possibilities this new technology offers need to be explored and methods developed. In this paper, we explore the monitoring capabilities of virtual fencing technology on a herd of cattle in a coastal dune landscape. We explore to what extent a herd of cattle prefers some habitats over others, and in what way they utilise each of the available habitats. We find clear differences in the amount of time the herd spends in each habitat and in the ways they utilise each habitat. The herd spend a disproportionately large amount of time in salt meadow, and likely spend most of their time there resting and ruminating. We conclude that the method for monitoring of cattle applied in this study, using existing virtual fencing technology, is a relatively precise method useful in year-round monitoring, with room for improvement.

**Abstract:**

There has been an increased focus on new technologies to monitor habitat use and behaviour of cattle to develop a more sustainable livestock grazing system without compromising animal welfare. One of the currently used methods for monitoring cattle behaviour is tri-axial accelerometer data from systems such as virtual fencing technology or bespoke monitoring technology. Collection and transmission of high-frequency accelerometer and GNSS data is a major energy cost, and quickly drains the battery in contemporary virtual fencing systems, making it unsuitable for long-term monitoring. In this paper, we explore the possibility of determining habitat preference and habitat utilisation patterns in cattle using low-frequency activity and location data. We achieve this by (1) calculating habitat selection ratios, (2) determining daily activity patterns, and (3) based on those, inferring grazing and resting sites in a group of cattle wearing virtual fencing collars in a coastal setting with grey, wooded, and decalcified dunes, humid dune slacks, and salt meadows. We found that GNSS data, and a measure of activity, combined with accurate mapping of habitats can be an effective tool in assessing habitat preference. The animals preferred salt meadows over the other habitats, with wooded dunes and humid dune slacks being the least preferred. We were able to identify daily patterns in activity. By comparing general trends in activity levels to the existing literature, and using a Gaussian mixture model, it was possible to infer resting and grazing behaviour in the different habitats. According to our inference of behaviour the herd predominantly used the salt meadows for resting and ruminating. The approach used in this study allowed us to use GNSS location data and activity data and combine it with accurate habitat mapping to assess habitat preference and habitat utilisation patterns, which can be an important tool for guiding management decisions.

## 1. Introduction

Recent advances in precision livestock farming technologies, such as GNSS-based virtual fencing, allow for greater control of grazing pastures and easier monitoring of animals, providing potential benefits for both production, nature conservation, and animal welfare [1,2,3]. This is especially true in areas with sloped and hilly terrain, where traditional physical fencing can be challenging and labour intensive [4,5]. In the last few decades, there has been an increased focus on monitoring livestock in both production and nature conservation settings to make better and more informed management decisions [6,7,8,9]. Especially, the habitat use and behaviour of cattle in regards to environmental concerns and animal welfare are of high interest [7,9,10,11]. Understanding habitat preferences of grazing livestock and their use of grazing areas is important to help develop a more sustainable livestock grazing system with minimal negative impact on the environment [11]. Multiple studies have investigated habitat use by free-ranging livestock to make better management decisions and promote resource conservation [12,13,14,15]. Cattle are generally less selective in their feed preferences compared to other grazing livestock such as sheep and goats [16]. They prefer habitats with a high biomass production such as meadows and grasslands but are also used in nature conservation on salt meadows and heathland with lower biomass production [15,17]. However, because grazing resources are often spatially and temporally heterogeneously distributed, animals need to visit different habitats to satisfy their nutritional needs [13]. In addition, terrain characteristics such as the steepness of slopes and the distance to drinking water also affect the habitat selection of cattle [13,14,15]. The habitat distribution of cattle is furthermore affected by temperature, protection from pests, and shelter from the elements [18]. GNSS-based systems with built-in accelerometers can be used to monitor both spatial distribution of animals and their movement, which can potentially allow monitoring of animal behaviour and activity [19]. The collected data can be used to monitor variations in daily animal activity and help characterise typical behavioural patterns, which make it feasible to detect deviations when they occur, thereby making it possible to detect potential disease and/or welfare concerns remotely [11,20]. As such, development of methods to remotely classify and monitor behaviour of livestock from data collected by technologies such as virtual fencing systems could both improve the value of such technologies and provide welfare improvements for livestock, as health and welfare concerns could be detected rapidly.

One of the currently used methods for monitoring activity and classifying behaviour of livestock is tri-axial accelerometer data [19,21,22]. An example of this method can be found in a recent paper by Versluijs et al. [19], where accelerometer data from a virtual fencing system were used to accurately classify the behaviour of beef cattle in a semi-natural forest setting. This method, although displaying high fidelity in classifying behaviour over shorter time frames, quickly drains the system of power with its high data communication demands and is, therefore, not usable when considering constant long-term monitoring. This limits research to short consecutive periods [19]. To enable longer-term studies with continuous data and constant year-round monitoring a method needs to be developed based on far simpler and less power-consuming data communication demands. One previous study has used only GNSS data to classify behaviour based on threshold values for movement velocity, calculated as distance travelled in a period of time [23]. The study by McIntosh et al. [23] uses GNSS location data collected at 5 min intervals, which according to Augustine and Derner [24] cannot provide misclassification rates lower than 12 to 16%. If a lower error rate is desired, Augustine and Derner [24] propose combining GNSS data with other types of sensors. The standard interval for GNSS location determination by a virtual fencing system developed by Nofence*©* is 15 min. This interval is significantly longer than 5 min, and as such, additional sensor information is likely necessary to classify behaviours. Incidentally, the Nofence*©* system provides a measure of summed activity every 30 min. In this paper, we explore the possibility of determining habitat preference and habitat utilisation patterns using low-frequency location data and a coarse measure of activity. Habitat preference is defined by Matthiopoulos et al. [25] as: The ratio of habitat usage over its availability. Habitat utilisation patterns are loosely defined by us as: At what time of day and in which way the animals use each habitat, e.g., do the animals use a habitat mainly for grazing or for resting. We achieve this by (1) calculating habitat selection ratios, (2) determining daily activity patterns, and (3) based on those, inferring grazing and resting sites in a group of grazing cattle wearing virtual fencing collars. All three of these criteria must be met for the method to be considered successful.

## 2. Materials and Method

All data in this study were provided by ‘Projekt Virtuelt Hegn, Fanø’ www.virtuelthegn.dk accessed on 2 October 2023). The management of the enclosure and the animals was carried out entirely by the farmer at ‘Projekt Virtuelt Hegn, Fanø’ for this study.

### 2.1. Animals and Location

This study took place on the western coast of the Danish island of Fanø, located in the southwestern part of Denmark in the Wadden Sea (Figure 1). The size of the study area was 163.5 hectares and the area consisted of a mosaic of different coastal habitats. The initial mapping of habitats was carried out by downloading existing mapping by the Danish “National Monitoring and Assessment Programme for the Aquatic and Terrestrial Environment (NOVANA)” from www.arealdata.miljoeportal.dk (Kortlægning af naturtyper—flader). The most recent mapping performed by the NOVANA program in the area is based on in-field observations and assessments by trained field biologists in the years 2016–2018. The mapping of habitats by the NOVANA program was supplemented and lightly modified by us using elevation models and aerial photographs (Dataforsyningen.dk (Forårsbilleder Ortofoto—GeoDanmark); orthophotos from spring 2022). The modifications made were merging of different habitat subtypes with similar vegetation compositions; mapping of wooded areas, as these were not covered by the NOVANA program; and minor alterations to the existing NOVANA mapping due to observed differences in mapping and actual conditions. Observed differences were likely due to the age of mapping, as some areas were last visited by the NOVANA program in 2016. The mapped habitats in the enclosure, with characteristic vegetation as reported by the NOVANA program, area, and percentage of total area are listed in Table 1, and the geographic extent shown in Figure 1.

The study animals were 17 Angus beef cattle (*Bos taurus*), of which 16 were heifers, 10–17 months of age, and 1 was an older cow, 11 years of age. All animals were non-pregnant. Their precise weights were not known at the time of this study. All animals had been at the study location and within the virtual enclosure for at least two months before data collection for this study began. The cattle were enclosed using a virtual fencing system developed by Nofence*©* (Molde, Norway). Each animal was fitted with a Nofence*©* collar capable of logging the location and activity of the animal, as described by Aaser et al. [28]. The size of the enclosure had gradually increased over two months from the initial introduction of the animals to the study area, but was kept constant at 163.5 hectares from July 11 onwards. To ensure that no changes in the size or extent of the enclosure occurred during the period analysed, only data from July 11 until the time of download of data on October 2 were included in this study. The data used in this study were downloaded from the database provided to us by Projekt Virtuelt Hegn, Fanø [29]. For the duration of the data collection period, an average of 3.79 electrical impulses and 47.93 audio warnings were recorded per day for the whole herd. A single animal “escaped” the enclosure once, but stayed only just outside the virtual enclosure for approximately 40 min, before returning to the enclosure on its own initiative.

### 2.2. Data and Statistical Analysis

The collars sent two separate types of data. One type contained just the position of the animal and was sent every 15 min. The other type was sent every 30 min and contained a measure of activity, henceforth “activity index”, as well as the position of the animal at the time of sending. The activity index was given as a converted measure of movement, as registered by the built-in accelerometer. The collars contained a tri-axial accelerometer, as they were the same model as the ones used by Versluijs et al. [19], but with a slightly different data output. Access to the raw tri-axial accelerometer data requires custom-made firmware, which was unavailable for this study. As such, the obtained activity index value from the collars was a sum of the registered movement since the last activity data point was sent; that is, every activity index described the movement of the collar in the 30 min leading up to the sending. The exact method of converting the raw accelerometer data to an activity index has not been disclosed by Nofence*©*. The values were not capped, and as such could vary from zero to an indefinite number. Since the software converting the raw accelerometer data to the activity index was originally developed for sheep, the measure was given as a unitless number with no biological meaning.

All recorded data points were assigned a habitat based on their position within the virtual enclosure (see Figure 1). Points that fell outside the virtual enclosure, either due to GNSS inaccuracy or the animal having momentarily escaping the enclosure, were excluded from the data. In this cleaning of the data, 1398 points (0.7% of the dataset) were removed due to falling outside the virtual enclosure, 3 of which stemmed from the single recorded “escape”. Along with the position of the animal, the collar transmitted the horizontal accuracy of the GNSS receiver. Exactly how this horizontal accuracy was determined was not disclosed by Nofence*©*. From this information, the average horizontal accuracy of the GNSS data points was 7.7 m with a standard deviation of 3.9 m. The activity index values were inspected visually and determined to display clear bi-modality. This bi-modality stems from the daily activity pattern of cattle, with distinct periods of high activity behaviours, such as grazing, separated by periods of low activity, usually due to resting [30,31,32]. Therefore, all activity index values were classified as either high or low activity (Figure 2). This was achieved using a Gaussian mixture model on the log-transformed activity index values. Before log-transformation, 1 was added to all activity index values to avoid the problem of log-transforming values of zero. The threshold value for classifying an activity index value as high rather than low was 764.

Habitat preference was analysed using the position-only data points, while the analysis of behaviour and activity patterns was based on the activity index data. The analysis of habitat preference consisted of calculating habitat selection ratios for each individual cow [33]. The habitat selection ratio was defined as the proportion of time an animal spends in a habitat relative to the availability of that habitat [33,34]. The selection ratio (*SR*) was calculated as
$$SR=\frac{\frac{n_{i,x}}{N_{i}}}{\frac{a_{x}}{A}}$$
where $n_{i}$ is the number of position-only data points logged by cow *i* in habitat *x*, $N_{i}$ is the the total number of position-only data points logged by cow *i*, $a_{x}$ is the area of habitat *x*, and *A* is the total area of the enclosure. In this way, a selection ratio above 1 indicates a preference for that habitat, while a selection ratio below 1 indicates preference against the habitat. To test whether the observed preference for or against each habitat deviated significantly from a null hypothesis of no habitat preference, a $\chi^{2}$ test was used. The $\chi^{2}$ statistic was calculated for each combination of individual cow and habitat, and then, added together by habitat. The sum of the 17 $\chi^{2}$ statistics for each habitat was used for the $\chi^{2}$ test (with 16 degrees of freedom), as recommended by Manly et al. [33] and White and Garrott [35] in Calenge and Dufour [36]. Additionally, possible differences in selection ratios between habitats were tested using pairwise Mann–Whitney *U* tests. The analysis of activity and habitat utilisation patterns was primarily performed qualitatively, by visual inspection of the activity index, with all data pooled, irrespective of individual.

Mapping of habitats was performed in QGIS version 3.26 [37], and all statistical analyses were performed in R version 4.3.0 [38].

## 3. Results

### 3.1. Habitat Preference

All habitats were found to be significantly selected for or against using $\chi^{2}$ tests. All $\chi^{2}$ tests yielded a p<0.001 at 16 degrees of freedom, with the $\chi^{2}$ statistic ranging from 3055 to 1,558,201. Pairwise Mann–Whitney *U* tests on habitat selection ratios, showed that animals preferred salt meadows significantly more than all other habitats (p<0.001). As such, the selection ratio for salt meadows was 6.78 ± 0.36 (reported as median ± median absolute deviance (MAD)) (Table 2). There was a significant difference in habitat selection ratios between all habitats except between wooded dunes and humid dune slacks (Figure 3).

Out of the five habitats in the enclosure, two were selected for and three were selected against (Figure 3). The habitats that were selected for were salt meadows 6.78 ± 0.36 (median ± MAD) and grey dunes 1.32 ± 0.04 (median ± MAD). The three habitats selected against were decalcified dunes 0.75 ± 0.05 (median ± MAD), humid dune slacks 0.52 ± 0.03 (median ± MAD), and wooded dunes 0.47 ± 0.04 (median ± MAD) (Figure 3).

### 3.2. Daily Activity Patterns and Behaviour Classification

The activity of the animals was found to display a clear daily rhythm, with median activity index values being far higher in the early morning and late afternoon/evening than during midday and night hours (Figure 4A). The same pattern was found in the proportion between high and low activity index values throughout the day, with high activity index values being far more prevalent around dawn and dusk, and low activity index values dominating around midday and during the night (Figure 4B). Overall, the animals spent significantly more time at low activity levels (median of 13 h/day) than at high activity levels (median of 10.5 h/day). The two medians do not quite add up to 24 h. This was likely due to the first and last day of the study period not being complete days, and slight inconsistencies in the collars, meaning sometimes only 47 (and not the expected 48) daily activity measurements would be recorded. These inconsistencies seemed to be irrespective of collar and to occur randomly.

The behaviour (activity classes) and location of the animals were not validated with in-field observations in this study. Nevertheless, we believe the classification of activity index values as either high or low activity provided a good indication of the behaviour of the animals. Therefore, analysis of habitat utilisation was performed by inferring grazing and resting sites from the activity index values in Figure 4A,B, compared with a qualitative visual inspection of habitat preference by time of day (Figure 5).

In decalcified dunes, there were relatively more points recorded during low-activity hours, especially at night, than during high-activity hours (Figure 5, decalcified dunes). In grey dunes, there were more recorded points during the daytime than at night, where the activity index value was lower (Figure 5, grey dunes). In humid dune slacks, there was a clear pattern of more recorded points during periods of high activity in the morning hours, late afternoon, and early evening. Consequently, humid dune slacks had clear dips in the number of points recorded during low activity hours at night and around noon (Figure 5, humid dune slacks). Salt meadows likewise showed a clear, but opposite, pattern. The number of recorded points in salt meadows was higher during periods of lower activity, with more points recorded during the night and around noon, and fewer around periods of high activity, such as in the morning, late afternoon, and early evening (Figure 5, salt meadows). Not a lot of time was spent in wooded dunes, but there was a clear pattern of more recorded points during the night, in periods with the lowest activity (Figure 5, wooded dunes).

## 4. Discussion

By combining GNSS data from a group of cattle wearing virtual fencing collars and mapping of habitats from publicly available field data and aerial photos, we successfully mapped habitat preference of a herd of cattle within a virtual enclosure. This method has previously been shown to be an effective way of assessing habitat preference as an estimate of grazing pressure [39]. In the present study, the herd showed a clear preference for salt meadows, as evidenced by the significantly higher selection ratio for this habitat 6.78 ± 0.36 (median ± MAD) compared to all other habitats 1.32 ± 0.04 (median ± MAD) to 0.47 ± 0.04 (median ± MAD). This preference for salt meadows is not immediately logical according to the existing literature, as cattle generally prefer grazing on drier habitats, as these tend to have vegetation with higher crude protein and lower fibre content than wet habitats [11,40], although one study has found cattle to preferentially graze wet areas [41]. In a previous study in a mosaic landscape of sand dunes and lowland habitats, somewhat comparable to this area, cattle also preferred grazing in the lowland habitats [42]. This seemingly counter-intuitive preference could be a result of what other habitats are available. Across several studies, the least preferred vegetation for grazing by cattle is half-shrubs, such as heather (*Calluna vulgaris*), irrespective of cattle breed and season [39,43,44]. This would explain the preference for salt meadows over both decalcified dunes and grey dunes, which are usually characterised by heather [45]. Likewise, humid dune slacks are generally dominated by sedges, which cattle also tend to avoid when other options are available [40,45]. Another explanation for the preference for salt meadows over other habitats is the productivity of the habitat. Studies suggest that cattle prefer habitats characterised by high biomass production [15,39,44], and of the five classified habitats in the study area, salt meadows have the highest biomass production and highest nutritional value for cattle [17,46]. The least preferred habitat in this study was wooded dunes, with a selection ratio of 0.47 ± 0.04 (median ± MAD). This is in line with previous studies that have found cattle to avoid wooded areas and have higher occupancy of open grassland in both extensive and intensive grazing conditions [12,47,48]. The narrow range in habitat preferences found between individual animals indicates a quite cohesive herd structure, with all animals likely being in the same habitat at the same time (Figure 3). Habitat preference could possibly have shifted over time during this study, but this was not examined in this study.

Despite not being able to directly quantify habitat utilisation in this study, we were able to identify some general trends based on patterns in activity and differences in habitat selection ratios. Quantifying habitat utilisation would require classification of behaviours, as performed by others, such as Ungar et al. [6] and Versluijs et al. [19]. The methods used by Versluijs et al. and Ungar et al. both require ground-truthing, i.e., in-field observations of the animals for validation of the behaviours inferred from the activity measures. Alternatively, behaviour classification has also been shown to be possible using GNSS data only, although collected at higher frequencies than in the present study [23,24]. Whether or not any of these other methodologies could perform equally well or better than the methodology used here, using the same dataset, was not explored to keep this study simple. Further research in this area could improve the understanding of the potential of using readily available data from virtual fencing systems.

In accordance with the existing literature, we found that cattle spend a majority of the day at activity index values categorised as low activity levels (median of 13 h/day) [32,49], and that cattle exhibit periods of activity index values categorised as high activity during early morning and late afternoon, with periods of low activity in between (Figure 4) [31]. The high level of variation in activity index values during daytime hours is most likely due to the combined effects of weather and season (Figure 4) [30,31,32]. The effect of season is especially pertinent in this study, as the length of day has been shown to significantly affect the activity and behaviour of cattle [30,31,32]. The period of data collection was characterised by a shortening of the day from around 17 h of daylight to 10.5 h (source: WorldData.info; retrieved on 19 December 2023). Based on previous studies, we can infer that a low activity level likely covers behaviours such as resting and ruminating, while high activity is a sign of grazing and/or walking [6,32,49,50]. These inferences indicate that while the herd of cattle preferentially stayed on the salt meadows, they might have predominantly used the area for resting and ruminating rather than grazing, as most of the time spent on salt meadows was during periods of generally low activity (Figure 5, salt meadows). Previous studies have shown cattle to prefer resting and ruminating near water sources and on nutrient-rich vegetation [11,51]. However, cattle also seem to prefer grazing near water sources [11,41], which could explain why most instances of being in humid dune slacks were recorded during high-activity periods (Figure 5, humid dune slacks). Humid dune slacks were one of the least preferred habitats (23.1 ± 1.49 points/ha), and the majority of grazing likely did not take place there.

Climate and weather plays a major role in explaining animal behaviour and habitat use [11]. The relative preferential use of decalcified dunes and wooded dunes during night hours, and grey dunes during hours of daylight, is likely due to the weather patterns of the study site. Due to the location, topography, and vegetation of the different habitats at this particular study site, decalcified dunes and wooded dunes would have provided the most cover against the prevailing winds at the study site, with grey dunes providing the least amount of cover. Wooded dunes would also have provided natural shelter against precipitation [11]. It is highly unlikely that much grazing happened in the decalcified and wooded dunes, as the majority of the time spent there by the cattle was at low activity levels and during night hours, when cattle have been shown to avoid grazing (Figure 5) [40].

Our results showed that GNSS data and a coarse measure of activity, given as an activity index, combined with accurate mapping of habitats can be an effective tool in assessing habitat preference and general trends in habitat utilisation. Additionally, by utilising existing technology integrated in virtual fencing systems, this is an effective method of monitoring cattle in extensive settings, without the need for additional sensors, which also increases the value of virtual fencing technology [11]. This type of method for monitoring the behaviour of cattle could also provide animal welfare benefits, as health and welfare issues could potentially be detected in almost real time if the system becomes automated and an animal’s behaviour is found to change significantly. For example, such a change could be a clear and sudden shift in the proportion of time spent grazing or resting. One potential easy improvement to the method of this study is to include the distance the animal has travelled between two activity recordings. This simple addition would likely allow for more accurate behaviour classification, as performed by Ungar et al. [6] and Ganskopp et al. [52]. Additionally, by measuring the distance the animal has moved between two points and the time between those two points, it would be possible to calculate an average speed of movement that could also be used for classification of behaviours, as performed by McIntosh et al. [23], inspired by Augustine and Derner [24]. As such, simply by expanding the use of the readily available GNSS data from the virtual fencing collars, the method used in this study could be improved. Another potential improvement is recording activity counts in two dimensions (fore–aft and left–right). Modelling behaviour on two-dimensional accelerations rather than on one-dimensional activity measures allows for greater fidelity in behaviour classification and is a widely used method [6,9,50]. Alternatively, using tri-axial accelerometer data provides the highest fidelity in classifying behaviour but also requires the highest frequency of data collection [19]. Collecting tri-axial accelerometer data and classifying behaviours by modelling is possible using the same virtual fencing system as used in this study, but requires purpose-made firmware and quickly drains the system of power due to the high frequency of data collection and transmission required [19]. However, by simply transmitting the raw pre-processed data from the built-in accelerometer instead of the converted unitless activity index used in the current study, behavioural modelling might be possible without increasing the frequency of data collection and transmission. We believe that although the method used in this study does not allow for fine detail analysis, it does provide some general insights into habitat use and preference of cattle that can be useful for management decisions. Although the current study was limited to around three months of data collection, our method allows for year-round studies. This is important, as habitat use of cattle has been shown to be season dependent [11,13].

## 5. Conclusions

The method in this study was successfully used to map habitat preference and general trends in activity patterns of a herd of cattle in a mosaic dune landscape. Although we did not carry out in-field observations to validate our behavioural classification, we were able to infer grazing and resting sites in different habitats and at different times of day by comparing general trends in activity levels to the existing literature. In this study, the herd of cattle had a significant preference for salt meadows, while wooded dunes and humid dune slacks were the least preferred habitats. The herd had a clear diurnal activity pattern with two distinct periods of high activity: one period in the early morning and one in the late afternoon. According to our inference of habitat utilisation, the herd predominantly used the salt meadows for resting and ruminating rather than grazing. The differential preference for use of the habitats present in the study area, could likely be explained by differences in vegetation cover, topography, and moisture. Although our developed method could be improved upon, we believe the use of GNSS location data and a coarse measure of activity, given as an activity index, combined with accurate mapping of habitats is an effective tool for collecting site and livestock data and interpretations that could not be collected cost-effectively using manual labour on foot. This information could help monitor and detect health and welfare concerns of cattle rapidly. By utilising existing technology and data from already implemented virtual fencing technology, this allows for long-term year-round studies without the need for purpose-made systems. A small amount of extra data processing and potentially improved communication of data from the existing virtual fencing technology, could improve the ability to accurately map habitat utilisation and classify behaviours, thus increasing the value of virtual fencing technologies.

## Figures and Tables

**Figure 1 animals-14-01506-f001:**
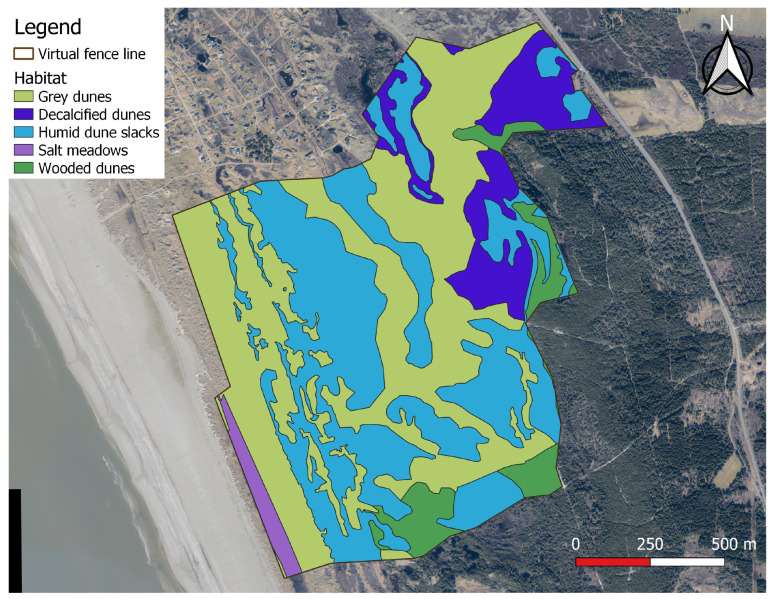
The study area and mapped habitats: each colour represents a different habitat, with the brown line outlining the virtual fence line in the study period (11 July 2023 to 2 October 2023). Contains data from The Danish Agency for Data Supply and Infrastructure [26]. Contains data used in accordance with the terms of use of Danish public data [27]. The coordinates of the upper left corner of the map are 8°23′01″ E 55°24′37″ N.

**Figure 2 animals-14-01506-f002:**
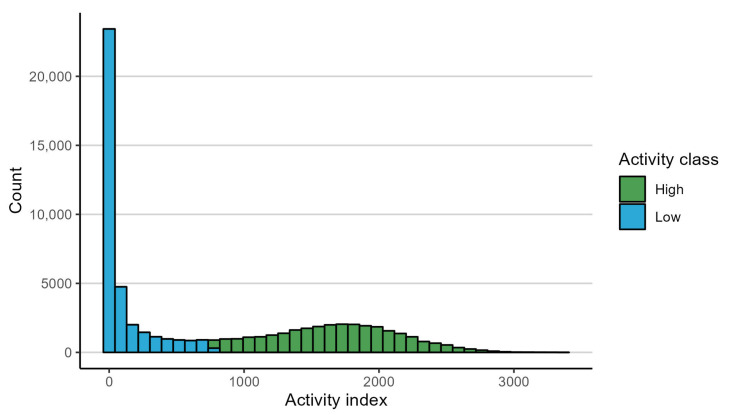
Recorded activity index values for all cattle combined. The bars are coloured based on classification as either high or low activity. One bar is mixed due to the threshold value (764) being within a bin rather than constituting a bin edge.

**Figure 3 animals-14-01506-f003:**
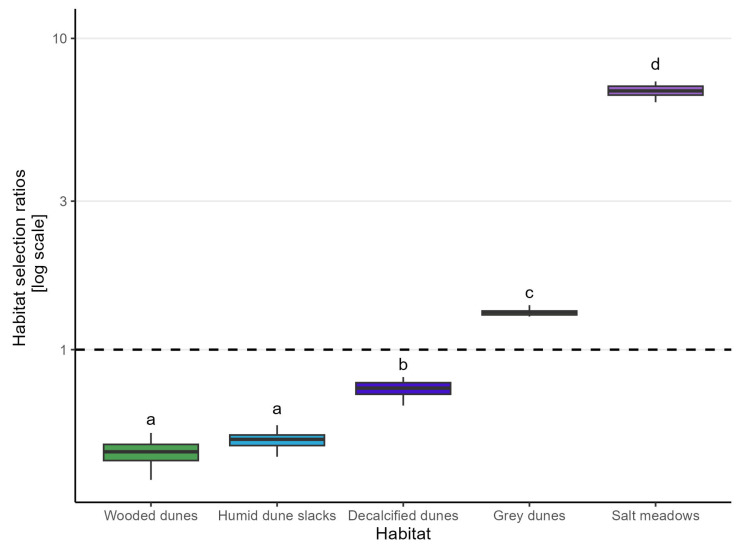
Habitat selection ratios calculated as proportion of recorded position-only points in a habitat divided by the availability of the habitat, for each individual animal. For each box, *n* = 17. The boxes show the median habitat selection ratio of the 17 animals, as well as the first and third quartile. The whiskers extend to the most extreme point within 1.5 times the inter-quartile range above and below the third and first quartiles, respectively. Only three selection ratios in total were outside these ranges and are not shown here to improve legibility. The dotted line indicates a proportion of time spent in a habitat equal to its availability, i.e., random use relative to availability. Therefore, values above the dotted line indicate that the cattle actively select for the habitat, while values below indicate that the cattle actively select against the habitat. The y-scale has been log base 10-transformed. Different letters indicate statistical significant difference with pairwise Mann–Whitney *U* test after Bonferroni correction; $p_{bonf}<0.01$.

**Figure 4 animals-14-01506-f004:**
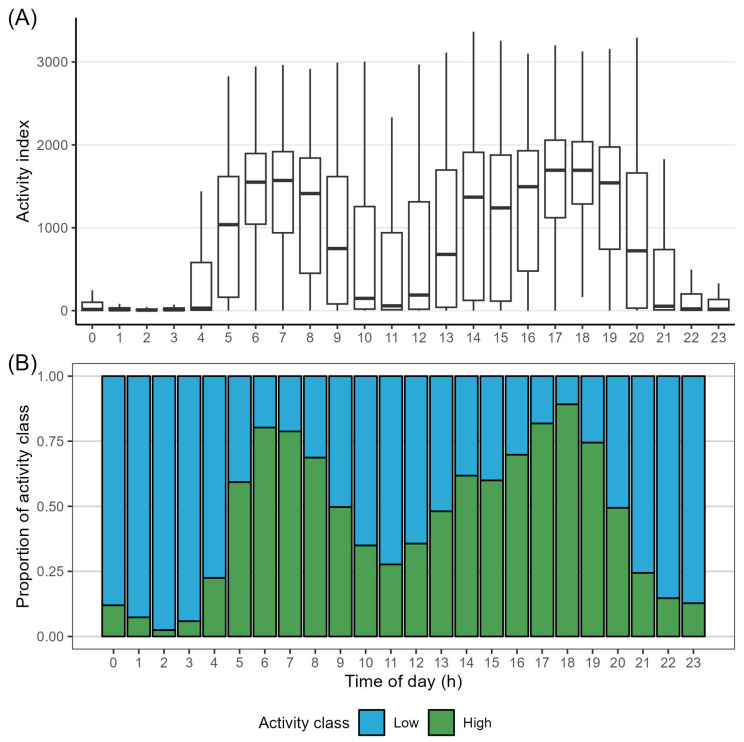
Daily activity patterns, with (**A**) showing the distribution of all recorded activity index values for every hour of the day. Outliers are not shown to improve legibility. (**B**) shows the proportion of low and high activity index values for every hour of the day. Sunrise and sunset were at 04:59 and 22:04, respectively, on the first day of data collection and at 07:30 at 19:01 on the last day of data collection.

**Figure 5 animals-14-01506-f005:**
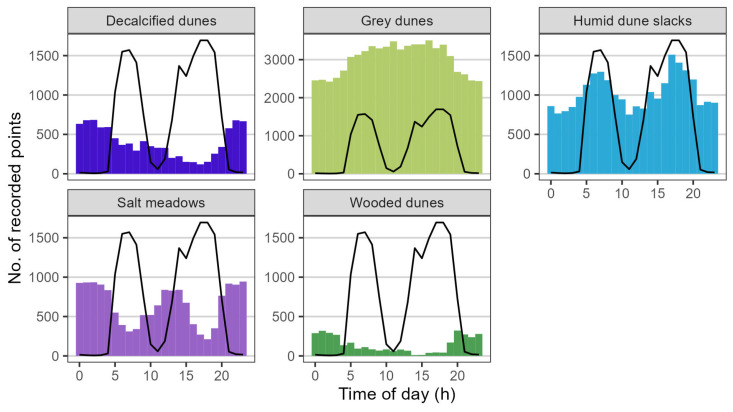
Habitat use by time of day, visualised as the number of location-only points recorded in each habitat for every hour of a day. The data are from the entire herd for the full data collection period. The black line indicates the median activity index value for the corresponding time of day. This line corresponds to the median values of activity index shown in Figure 4A. Note that the y-axis for grey dunes was expanded to fit all points.

**Table 1 animals-14-01506-t001:** Overview of the habitats available in the enclosure. For each habitat the characteristic species found in the area are listed, as well as the available area of the habitat and the percentage of the overall area covered by that habitat.

Habitat	Characteristic Species	Area (ha)	% of Total Area
Salt meadow	*Phragmites australis, Carex extensa, Carex distans, Plantago maritima, Triglochin maritima, Agrostis stolonifera*	2.9	1.8%
Wooded dunes	*Pinus mugo, Pinus sylvestris, Pinus contorta, Picea sitchensis, Carex arenaria, Calluna vulgaris*	9.7	5.9%
Decalcified dunes	*Calluna vulgaris, Empetrum nigrum, Carex arenaria, Avenella flexuosa, Polypodium vulgare*	16.6	10.2%
Humid dune slacks	*Salix repens var. argentea, Equisetum fluviatile, Eriophorum angustifolium, Drosera intermedia, Gentiana pneumonanthe*	62.8	38.4%
Grey dunes	*Ammophila arenaria, Corynephorus canescens, Carex arenaria, Calluna vulgaris, Festuca ovina, Jasione montana, Potentilla erecta, Cladina* sp.	71.5	43.7%

**Table 2 animals-14-01506-t002:** Median selection ratio and median absolute deviance (MAD) for each habitat.

Habitat	Median Selection Ratio	MAD
Salt meadow	6.78	0.36
Grey dunes	1.32	0.04
Decalficied dunes	0.75	0.05
Humid dune slacks	0.52	0.03
Wooded dunes	0.47	0.04

## Data Availability

The data presented in this study are available on request from the corresponding author.

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
