# Peer review of "Using Activity Measures and GNSS Data from a Virtual Fencing System to Assess Habitat Preference and Habitat Utilisation Patterns in Cattle"

_animals, 2024, doi:10.3390/ani14101506_

Round 1

Reviewer 1 Report

Comments and Suggestions for Authors

Overview

This paper looks at location (and through this habitat presence/absence) and activity together from collar-based sensors in a small herd of cattle in a mixed natural setting. The collars used are actually Nofence (NF) virtual fence (VF) collars, though the virtual fencing element is not covered or discussed in this paper. The link with VF is through either saying that this location/activity information is a cheap/easy set of data as a sort of by-product of using the kit, or it provides  potential improved functionality/value for the VF system. Either way it is about the location and activity data. A case could be made that VF systems need to provide more (and accessible) data to ensure their net welfare benefits can be harvested by farmers and for benefit of the animals themselves.

There is no visual observation validation/calibration data, so the two sensor dataset of GNSS and a motion index sit alongside each other, as a weakness (there is no groundtuth validation) or a strength (it works well without the major effort of observational data).

There is relatively little about animal welfare, relevant to the special issues topic of ‘welfare’, but that does not detract from the content of paper as a self-standing story.

The paper is clear, well-written, with generally excellent English.

Its shows that simple, relatively ‘motion’ infrequent data, that is the current status quo for extensive livestock because of either or both power use and communication methods can provide useful understanding of cattle behaviour, especially in relation to habitat use in mixed semi-natural settings.

There is no attempt to look at between animal, or within-animal behavioural change to indicate any issues of welfare and potential for either, or both combined, of the data sources (GNSS and simple activity index) to act as a potential alert for management, and thus welfare, reasons. Again not an issue, but a sort of gap compared to the Special Issue topic area.

My main concern is over how the story of the two sensor data types; GNSS data, Accelerometer/motion index data and how they interact.   I deal with this in some detail during detailed review and then as a final discussion at the end of this review. I think it is essential that this issue is dealt with.

Detailed comments for review, correction or rebuttal

Title : GPS – I believe for scientific writing GNSS is the correct term, unless any GNSS device uses solely the US GPS satellites. Nofence uses the wider spectrum. The same throughout manuscript, GPS is what the general public use, but science should be more specific. A matter for editors if any debate.

L4 and L12 and elsewhere throughout article - ‘low cost or cheap’. The Nofence system, and the same for other commercial VF systems, is not described as low-cost by farmer users. There are Norwegian theses that show users put ‘cost’ as one of biggest issues. The approach proposed here is perhaps a way of getting ‘extra or better value’ from a system used for other reasons (the fencing functions), though the ‘activity’ function is clearly accessible for farming users, and some use it. The combined cost of hardware, of the monthly/annual subscription charges and the labour costs of putting collars on/off and checking it on the apps provided should not be dismissed. It may be low cost compared to historic tracker systems such as Lo-Tek used by conservation grazers and many grazing ecologist and wildlife scientists. I suggest a review of wording throughout and ‘better value or added value’ considered instead.

L10 omit ‘a’ salt meadow

L16-18 I am far from convinced that tri-axial accelerometers are the ‘most used’, certainly not in free-ranging cattle in semi-natural or rangeland as opposed to intensive and dairy system research, where heavily used for oestrus detection etc.

L18 ‘power’. Furthermore the particular issue with common tri-axial accelerometers is that they provide most typically  12.5 (at 12.5 Hz, though the NF is 10Hz) x 3 data points every second. This creates enormous problems of data communication, and data packet size. GNSS location, and satellite fixing, is still I believe the major ‘power’ problem for much GNSS collar livestock research. The in-built solar panels in Nofence (and other VF systems – except US based Vence).  Something better might be  ‘Both tri-axial accels and GNSS sensor based systems have issues with power use, data packets sizes, and communication/transmission issues  for real-time monitoring. These are largely overcome in the NF system, through a solar-powered battery system, and providing relative infrequent data points (every 15-30 mins) providing condensed or summarised data only”. I believe the energy cost of tri-axial accelerometer data acquisition and writing to say a memory card is much less than the equivalent data collection of GNSS data, but the energy cost of the transmission of data by mobile phone system as per the Versluijs paper is what ‘kills’ the battery.  I agree the NF system needfully provides only simplified/condensed data, communicated with relatively low frequency (15-30 mins) to ensure power economics and long-term viability without battery changes. The challenge is how to get this across in a few words in an abstract (and later in the full text)

L26-27 You also inferred activity levels from the activity motion index. So ‘compared to literature’ alone seems maybe a rather too limited validation of your methods? See my note on the same phrase used in Conclusions. I think ‘somewhat’ is too weak and either nothing or perhaps ‘broadly’ is a better Conclusion/Abstract statement.

L30 ‘..allowed us to use GPS location for measuring the activity…’ I don’t believe you did this. Certainly not with GNSS alone.

Later, I will ask for coverage/discussion of literature that uses sequential locations to provide behavioural inference from GNSS data alone. Speed of movement/distances between consecutive locations has been used to crudely classify behaviour.

L31 ‘cheap’ again. The sentence is just as valid with ‘effective ‘ only?

L53 I would remove ‘all’ , it works as well without it, and it is unlikely that this habitat met all the cattle’s nutritional ‘wishes’. ‘….aiming to satisfy their nutritional needs’ might cover it better? Livestock clearly need to move to find other feeding areas as some diminish/change in nutritional qualities due to phenology and need to avoid over-exploiting what may be perceived as the best, and need to explore in the hope/prospect of finding better locations. They also need to do this as a herd. The paper is not about foraging theories, its just about ‘can we measure it..?’

L64 ‘most used’ again. I don’t think you need evidence of scale of use of tech in a variation of grazing contexts to make your point. Accelerometers alone have been used a lot, but mainly intensive, without location ndata. The Versluijs  paper is indeed one where they aimed to harvest behavioural classification (also from Nofence) in very extensive systems, using raw data rather than summarised and condensed, but one of a few that I know of. I believe data communication challenges, including its power use, rather than power use for the sensors themselves is the key technical challenge here.

At Line 68 I would suggest adding some data comms wording . For e.g.   ‘…shorter time frames, but with its high data communication demands, quickly drains the system…..’ or similar might be better.

L67 I would also suggest change ‘natural setting’ to something more specific clearer, such as semi-natural or mixed habitats etc. I don’t believe livestock specialists would readily be sure what ‘natural setting’ means.

Introduction section somewhere. Using GNSS data to ascribe behavioural characterisation. A number of authors have attempted to use GNSS data alone to create behavioural characterisation, particularly separating inactive, resting behaviour from active behaviour with walking, running and grazing. Half a sentence and the reference to Ganskopp mentions some of this in Discussion, but that is all. .  

McIntosh et al., 2022 is a more recent example for rangeland cattle grazing with a ecology paper Sarah Saldanha; Sam L Cox; Teresa Militão; Jacob González-Solís 2023 . There are quite a lot of issues with this approach, linked to resolution of GNSS fixes and time intervals, but it is used with GNSS data alone to get activity and movement categorisation.  

It is important that as a minimum you recognise this in the Introduction (see my final comments for a broader discussion of issues).

Methods

Figure 1. Initially when I looked at this, there appeared to be two colours of light blue, dark blue in the humid dune slacks areas. The underlying greyscale map, showing darker vegetation/forestry at time of the satellite image created initial confusion. I could be forestry or it could be stands of mature heather. You might want to make a note of this issues in the legend, or find a technical solution such as making the coloured map non-opaque to solve the issue.

Either in text or in this figure I think some map references, latitude/longitude should be given.

L105 It would be useful to add the average age and weights of the heifers (heifers is a wide category especially if deliberately un-mated) and note that all were non-pregnant, if they were. Were they purebred Angus or Angus crossbreds?

L108 would this not be better to say ‘fitted with a Nofence collar , capable of logging…’ to avoid potential confusion to readers about a possible second collar.

L115 You are not reporting the Nofence system performance, that’s fine, but for those readers not familiar with is, and potentially suspicious it might have modified the behaviour of the cattle I think a couple of sentences here about any interaction with the VF system would seem important. Just number of sound and electrical alerts, pulses per day might be helpful to confirm no real interaction. As you will likely know the GNSS system on board each NF collar modifies its satellite acquisition characteristics when close to the virtual boundary, which is intended to ensure that the resolution of the GNSS fix is better when it needs to be. Given the size of your study area then interactions with the NF system is likely quite low. Low incidence of interactions, and no break-outs, quite simplistically supports the view  that the cattle were not under nutritional pressure and wanting to move to a new area with better forage.

L123 etc . Consider calling the data points Motion or Activity index , Motion Value or something or even Activity Measurement Value – this could help labelling of figures etc. ‘Activity’ alone seems rather too non-numerical, it is still a number, and its scaling in your context is important (you use it to split Low and High Activity levels).

L126 Strictly it is movement of the ‘collar unit’ not the animal. This would make sure readers realise it will pick up neck movements associated with a lying cow, when ruminating for example, but this movement could be less than say a grazing cow.

L134 I always have an issue with removing points outside the ‘enclosure’ because whilst this usually removes spurious fixes that does fall outside, it cannot remove equally spurious, but unknown, data when it is all inside the grazing area. Not suggesting you change anything. We, as a matter of interest have ‘cleaned’ up similar data by using altitude as a filter to remove some occasional GNSS locations where the fix was quite far from a known groundtruthed location and thus used this method for grazing livestock with confidence it removes some poor fixes wherever they occur. Reporting the number/proportion of fixes removed during data cleaning would be helpful here.   

This sentence also mentions ‘momentarily escaping virtual enclosure’ A numerically based  comment on incidence of this and the ‘outside vitual fence’ would reassure readers re the robustness of the main data set used.

Fix resolution, accuracy of the Nofence collars would be useful to quote at this point. Most authors looking at habitat use with GNSS would quote some statistics or a reference here. With your smaller habitat parcels e.g. the rows of humidified dune slacks and grey dunes nearest the beach that are particularly small and narrow, and that salt meadow strip, then the resolution and accuracy of the GNSS fix will be important to avoid unknown miss-characterisation to habitat zones.

Line 139 and Figure 2: your histogram has a mixed bar at c 800.  Not an issue but text could usefully to say what that actual threshold number was.

Figure 2 ; The  legend should not really be a statement of the result ‘i.e displaying clear bi-modality’ The figure legend and the x axis title saying a count of individual Activity measure datapoints / “Total count for all cattle combined” and Activity Index score values’ or similar . Please review all your behaviour Figures.

Para 148-160. Most of this sentence is in present tense (‘is’ etc) but L159,162,163  uses ‘was/were’. Please just check tense of Results throughout around consistency.

Somewhere in Methods – missing info

You refer later to the quite large shifts in day/night between mid June and early October. Providing either or both sunrise/sunset, or dawn/dusk times for your Day 1 and your Day End would be very useful. This information could be added to your Figure 4 for example either graphically (as arrows etc ) with footnotes of as extra words in the Legend. 

L168 ratios

Figure 3: This looks a nice clear Figure. However, I then tried to quickly understand the data points/parameters in each box plots. The legend should state what each box/whisker plot displays – median or mean of each cow etc? this data is just 17 cattle data points for each habitat? The narrow range implies considerable consistency between individual cattle thus the herd being both very cohesive and likely very synchronised. This should be picked up Discussion.

L186 not ‘don’t’

L191 initially confused by the terminology ‘quantification of habitat utlisation’ both here and later .You are referring to some form of link to visually observation in the field (though not necessarily at this site, with these cattle etc), either to calibrate or validate or both the Activity measure, or the Low/high Activity classes, or a more thorough set of observations  of these cattle in this study in situ. I would prefer this sentence to be replaced by something more specific e.g. “Validation of both the behavioural (Activity) and locational data in this study with visual observations  was not carried out. Nevertheless…… and therefore analysis of habitat utlisation…”. I do think that the Figure 5 presentation is particularly compelling information and statements noting (in better words) that despite lacking independent validation of cattle behaviour and location, you believe the sensor data alone can give a good interpretation of what is going on.

Figure 5. This is an excellent graphic,but needs a clear labelling/legend. X axis title – is it per cow, whole herd, total for whole period, per day ? The black line uses the Activity units/scale from Figure 4, the same numerical scale as on x axis, but should really be labelled appropriately on x axis or in legend,. Referral back to Figure 4 could make this Figure more self-standing.

Results – Missing issues. There is a huge amount of data being collected in this study and only a small proportion is presented. The following points perhaps would benefit coverage in results or noting their omission in Discussion (“we note some finer scale within-period differences, but chose not to include here to keep the story simple [in better words]” or maybe you have plans for further publication.

Figure 3 appears to show extremely narrow distribution, variability for each cow at each habitat type. It would suggest to me that the herd was very cohesive, and likely their individual behaviour patterns were very synchronised. Despite lack of formal visual behavioural observations, this data result might be confirmed by the checks on animal health and welfare that would be mandatory. It does lead onto the point (for Discussion) that this is the behaviour of one herd, not of 17 independent cattle.

Shifts in habitat use over time. Many of the plants typical of the habitats would change over time, with phenological changes associated with the season and environment, but also potentially some changes due to grazing pressure on the most preferred habitats. The quite large shift in hours of daylight/nighttime might have also changed patterns quite a lot.  Your Figure 5 habitat use split for e.g. between say 2 weeks at start of study and 2 weeks at end of study might be quite revealing.

Similarly the Figure 4 data split according to early and late dates might be quite revealing. Both the change in habitat use, the changes in daily patterns of activity and your combined data (figure 5) could be highly revealing, and hopefully supportive of the sensitivity of your methods to pick up responses of the cattle to vegetation and day/night shifts, but also support the robustness of the methods. At the moment I am not saying you should put this new data/new figures in, but that the paper might be better if you did.

Some maps of classified behaviour, Low/High Activity by habitat and day/night might be interesting. Lengthening your paper, and adding to complexity but showing the value of the combined GNSS and Activity Index for identifying actual use of the land area and habitats, and I suspect the ‘tightness’ of the patterns. This might also show some of cow tracks likely in this type of mixed habitat and an issue of management and local land disturbance you have not covered and potentially of interest for management purposes. Just an idea if any further publication, or a more major re-write is required by editorial.

Discussion

L240 This is where you discuss the issue of having no direct validation, with observational data, of the activity index data. See my final discussion point below.

At L241 you say this would ‘require a higher frequency of data’. I would dispute this (the higher frequency element) , you just need contemporary observations alongside your actual sensor data – the Activity index every 30 minutes. Suggest simplifying this sentence and omitting that you would need different, more frequent sensor data. This sort of validation needs enough visual observations that match the actual sensor data used.

L279 You bring in again the lack of validation/calibration with visual observations, you use ‘quantification of habitat utilisation..’ which I do not like. It seems more appropriate to state that without validation, and without somewhat more categorises of behaviour (which probably needs more data per minute and perhaps combined data processing of both GNSS and accelerometers then the data interpretation is somewhat limited. And then as you already say (in abstract/Conclusions)..

L282-285. Using the GNSS data to interpret data such as speed, distance moved etc is somewhat a different point to using tri-axial accelerometers to collect, collate and transmit (you say ‘record’ which is not the issue with real-time systems, as the recording element will occur on a ‘cloud’ database). I say much elsewhere about GNSS data interpretation. The ‘getting more from Accelerometer data’ point is a good one nevertheless. There is a huge difference between the data transmitted by the NF collars in your study – one 1-3 digit number every 30 minutes versus the 1,000s transmitted in the Versluijs NF studies. Somewhere in between these cases seems to provide scope for a simple improvement, and indeed other farm management interpretation such as oestrus would require more than one number every 30 minutes, and likely more than one axis. But there is enormous scope with the same equipment. So your L286 statement ‘requires purpose made equipment’ seems odd. Using your own argument that you want to use NF system to provide more data, just needs to transmit more, and the most appropriate, pre-processed data from NF collars. it certainly does not need (as per your L286) a higher frequency of data collection (at the locus of the tri-axial sensor).

Sorry for the length of this point but basically I think you need 1-2 sentences on using  the GNSS data better, and 1-2 sentences on harvesting the already collected accelerometer data better (with a few more data points) and transmitting and making available (calculating on the collar, sending condensed, data) onto the NF app that data (that’s the recording element). You have already ‘sold’ the story that you want to create added value of data from NF collars on cattle when it is being used for other primary purposes, i.e. fencing (see I replaced ‘cheap’ with ‘better value’).

L298. Again I don’t like ‘unable to quantify habitat utilisation’. It seems you could be rather stronger on what you could say for example ‘Although we did not validate the sensor data with visual observations we were able to infer grazing…’. I suggest removing ‘somewhat’ perhaps you could replace with ‘broadly’ , ‘broadly’ means to me its about right, whilst ‘somewhat’ means its sometimes not right, something missing.

L308 ‘set goals’ ? what were your set goals? Where are they ? I did not remember them, so why here in Conclusions did you draw attention to apparent weakness in the study?

L310 ‘for guiding management decisions’ . I wonder what these management decisions might be in practice.   ‘…for collecting site and livestock data and interpretations that could not be collected cost-effectively using manual labour on foot’ or similar might be better.

L312 I would omit the ‘relatively infrequent..’ phrase. I am not sure what it adds. You could add ‘A small amount of extra processing and communication of data from the existing sensor systems could add more….etc..’ would seem a conclusion you could draw from your Para L274+

GNSS to behaviour linkages – main critique

I do think your Introduction and your Discussion needs to include more potential, strengths, weaknesses of using GNSS data alone etc. Certainly this needs to be better covered in the Introduction.

Then in  terms of your core Results I think you have a series of options

1)    Say you did not look further at this GNSS to behaviour/activity methodology because….

2)    Or better show/state results of comparisons between the two methods for your data set. This could be as little as a single (representative) day of comparison data. Either..

a.    This could show your Activity Index is better/good enough/simpler in computing terms etc than the current literature methods of GNSS to behaviour/activity

b.    Or act as a fuller sensor based validation for your Activity index method, which is the core of your combined GNSS location only/Activity Index methods. More of a cross-validation where both sensor data sets support each other interpretations.

Something covering Option 2) would I think strengthen the paper, especially if other reviewers/editor were not happy with lack of visual observation validation or calibration. Personally I think I would like a bit of 2 a) to confirm the large interval resolution of GNSS data alone was not that good and not needed and that the approach you have taken is enough.

But frankly at the moment, until/unless you do that  bit of comparison work we will not know which is better/complementary etc. This is the main reason for asking you to undertake this opportunity to finds out, and show how these two sensor approaches might fit together. It seems like a missed opportunity to not follow this through.

I have recommended 'Major' because this GNSS to 'behaviour' issue / opportunity needs dealing with in my opinion. And there is a clear opportunity to get more from the datasets, with a couple more Figures, tables to support the specificity and robustness of your approach to  garner more information on behaviour and welfare from Virtual Fencing collars. Without this issue undoubtedly a 'minor', just editing, paper.  

Recent references to GNSS to behaviour conversions..

Sarah Saldanha; Sam L Cox; Teresa Militão; Jacob González-Solís. 2023. “Animal Behaviour on the Move: The Use of Auxiliary Information and Semi-Supervision to Improve Behavioural Inferences from Hidden Markov Models Applied to GPS Tracking Datasets.” Movement Ecology 11 (41). https://doi.org/10.1186/s40462-023-00401-5.

McIntosh, M. M., Cibils, A. F., Estell, R. E., Gong, Q., Cao, H., Gonzalez, A. L., Nyamuryekung’e, S., & Spiegal, S. A. (2022). Can cattle geolocation data yield behavior-based criteria to inform precision grazing systems on rangeland? Livestock Science, 255. https://doi.org/10.1016/j.livsci.2021.104801

Reviewer 2 Report

Comments and Suggestions for Authors

This text investigates the integration of new technologies for monitoring cattle habitat use and behavior, aimed at fostering a more sustainable livestock grazing system that also upholds animal welfare standards. It critiques the prevalent method of monitoring, which relies on tri-axial accelerometer data from technologies such as virtual fencing. While informative, this method is criticized for its high power consumption due to the need for high-frequency data, rendering it impractical for prolonged use. The paper proposes an innovative approach that leverages low-frequency activity and location data to assess cattle's habitat preferences and utilization patterns, offering a potentially more sustainable and energy-efficient alternative.

Critically, the study highlights the application of this approach in guiding management decisions by providing a cost-effective and efficient means to monitor cattle behavior and habitat use through GPS location data and accurate habitat.

The methodology is structured around three primary activities: (1) quantifying cattle's preference for different habitats and comparing the proportion of time they spend in each habitat relative to its availability; (2) Analyzing the cattle's daily activity patterns to identify routine behaviors that might indicate habitat preference; and (3) pinpointing specific sites within the varied coastal habitats where cattle are most likely to graze and rest.

The ability to identify daily activity patterns and infer behavioral trends, such as grazing and resting preferences, underscores the potential of this method to offer a nuanced understanding of cattle behavior. This understanding is further enhanced by comparing these observed trends with existing literature, allowing for validating inferred behaviors against established knowledge.

I made special comments in the attached file.

Round 2

Reviewer 1 Report

Comments and Suggestions for Authors

Thanks for your new version. You have adequately covered my queries/comments in most cases, and the overall descriptions/detailing is now much better. I do like the story, that you have used a dataset with two sensor components to better understand what the cattle are doing etc... 

In terms of my main issue - the potential to use GNSS data alone to create some activity characterisation - now you have included more in Intro and Discussion to cover this, and made arguments that the accumulated, summed activity data is likely better than infrequent GNSS, then I am OK with your revisions here.

One small issue though - more about wording than substance. At your New Line 283 (in track change pdf). You say 'time constraints' in the paper, and make a similar, fuller, comment in your 'response document' . I fully understand that doing a whole new level of comparative analysis of a GNSS derive behaviour characterisation with your activity data sets was something you did not want to do, because it would require extra work, time etc. But for the academic paper saying effectively ‘we didn’t have enough time for analysis’ is not good scientific writing. Better to say ‘to keep things simple’ or ‘as we found the activity data alone convincing’  or similar arguments. Having thought about the issue a bit more, I would also foresee the problem that you could produce two different behavioural ‘pictures’ from GNSS only and from Activity index only, but without a third data set, such as some visual ground-truthing , the question would be, how would you know, how could you prove  which one is better?  They would just be different.

So I am happy you have taken the route you have.

A couple of other issues for Editor only to resolve

NL 160 ‘Horizontal accuracy was x meters etc. ‘ Thanks for adding this, but It is not clear how you derive this. Typically you need either repeated fixed locations, or some comparison with either a more accurate (e.g RTK GNSS) locations or other ground-truthed locations to create a data set to estimate horizontal accuracy. Or am I missing something?

L133 – thanks for adding number of data points removed during cleaning etc . Saying only “0.7% of data” etc is very helpful and supports strength of your data set.  Providing the number of sound alerts, the electrical pulses and escapes etc provides good context. At this point you give ‘average of 3.79 impulses etc’ . For added clarity, adding ‘for whole herd’ or similar would be good. Not that the detail is required but the strip of most preferred habitat (the salt meadow strip) is right on the western virtual fence boundary likely causes quite a lot of interaction with the VF boundaries, something you probably do not get in other boundary areas with less favoured vegetation.

Figure 3: I do not think you have dealt with my suggestion of clarity on the box and whisker figure. For other publications stating whether mean or median and enumerating percentiles etc is required. This is an editorial issue and I am not sure oin Animals policy.

For open-ness I will be saying to Editor, I am OK, except for these couple of issues above. But noting for handling Editor, as I did in Review 1, that the paper is ‘light’ on welfare, for a paper linked to a Welfare topic section. Personally, I am fine with paper as it is, but recommend consideration of moving it out of ‘Welfare’ into another topic etc. ort adding in more 'welfare' intro. and Discus..

Author Response

Thank you once again for excellent comments. We have made some final changes based on your feedback. We agree that using GNSS or Activity Index data only could result in two different results - this would be in interesting comparison in itself. The changes made are detailed below:

L133: Added “for the whole herd”.

L160: Added a short paragraph outlining where the reported accuracy of the GNSS positions stems from.

L283: We have changed “due to time constraints” to “to keep this study simple”.

Figure 3: Added a section in the legend explaining exactly what is shown on the figure, i.e. what the boxes and whiskers represent.